# Goal-Driven Human Motion Synthesis in Diverse Tasks

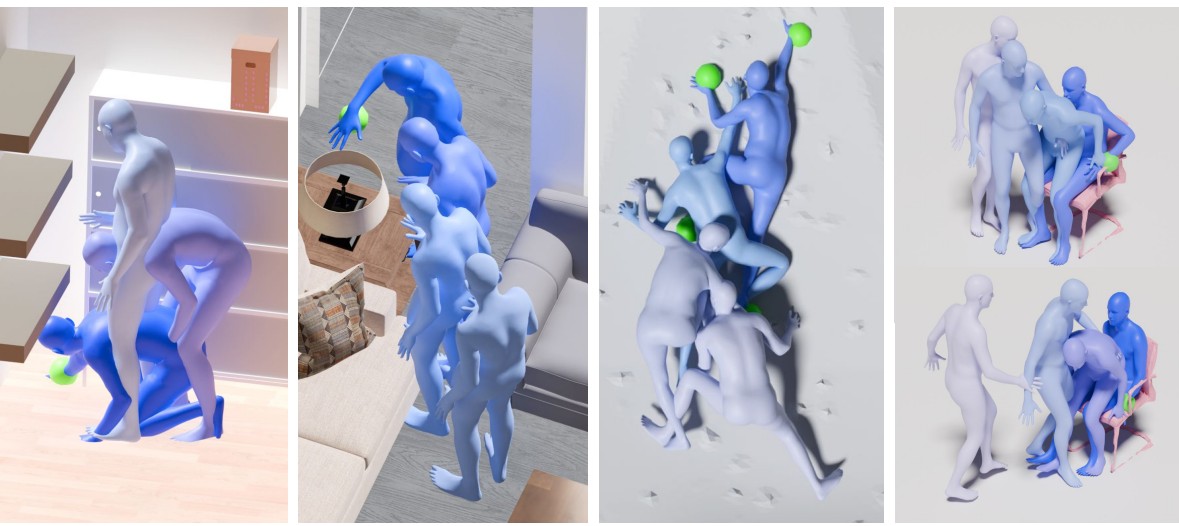

Figure 1. We propose a motion generation pipeline where pre-defined keyjoints approach user-specified positional goals. The goals are shown as green spheres, and our pipeline can adapt to the customized conditions including novel scenes and goal conditions. We can generate motions that reach for an object in cluttered scenes, climb a wall, or sit with specified hand positions.

## Abstract

*We propose a framework for goal-driven human motion generation, which can synthesize interaction-rich scenarios. Given the goal positions for key joints, our pipeline automatically generates natural full-body motion that approaches the target in cluttered environments. Our pipeline solves the complex constraints in a tractable formulation by disentangling the process of motion generation into two stages. The first stage computes the trajectory of the key joints like hands and feet to encourage the character to naturally approach the target position while avoiding possible physical violation. We demonstrate that diffusion-based guidance sampling can flexibly adapt to the local scene context while satisfying goal conditions. Then the subsequent second stage can easily generate plausible full-body motion that traverses the key joint trajectories. The proposed pipeline applies to various scenarios that have to concurrently account for 3D scene geometry and body joint configurations.*

## 1. Introduction

A goal-driven motion generation can streamline designing diverse interactive full-body motion. For example, when designing a character motion for grasping an item, setting a hand goal first allows user to efficiently formulate the desired functionality. Similarly, the users may describe the climbing motion by defining target positions or control the sitting posture by specifying contact points on a chair. In this paper, we propose a framework for generating natural full-body motion when the goal is simply the position of the key joints within a 3D scene. After a user intuitively defines the desired interactions by providing the target positions for the critical body parts, such as hands or feet, the system can generate natural full-body motion that is adaptive to the given condition.

Goal-driven motion requires satisfying part-wise goals while maintaining plausible full-body motion that is adaptive to unseen scene layouts. Such interaction is a highly challenging motion to generate. As the goals are defined on the input 3D scene, only a few existing captured motion data

precisely follow the required movement defined at the test time. We take inspiration from the recent advances in diffusion models, which have shown impressive performance in generative modeling, not only in image synthesis but also in human motion generation. These models learn continuous data distributions without collapsing and exhibit promising capabilities for control, such as compositionality [8, 33] or conditioning [66]. Another inspiration for enhanced control in diffusion models is guidance functions [4, 6, 47], which successfully endow customized properties into the outcomes via flexible sampling. We incorporate these techniques to formulate a diffusion model that generates motion approaching user-specified goals while avoiding collisions in diverse scenes.

We construct a two-stage diffusion model, solving simpler sub-problems to effectively tackle the overall complexity. We first generate a key joint trajectory that is adaptive to a customized goal position in a novel scene. Next, we generate natural full-body poses based on the predicted partial key joints. The key joint trajectories serve as an intermediate representation that detaches the complexity of scene perception and full-body generation. Both stages follow conditional diffusion formulation. The first stage employs a guidance function to sample the key joint trajectories that satisfy the goal conditions while preventing collisions. Here, our lightweight scene features provide the necessary spatial context, and the full body layouts are estimated as bounding boxes. The subsequent second stage composites the intricate full-body motion that matches the sampled trajectories of the partial key joints.

We demonstrate that our proposed method can accomplish the task even in unseen scenarios or newly defined goals without additional training. Our approach generally applies to a wide range of tasks, such as climbing or contact-designated sitting, where the precise control requirement is provided as goal positions for the key joints. In Fig. 1, we show various tasks that we could perform, with goals emphasized as colored spheres. In summary, our contributions are as follows.

- We propose a two-stage pipeline that efficiently generates motion that follows the goal positions of key joints while adapting to the target scene.
- We introduce an effective diffusion-based pipeline, which can generate plausible key joint trajectories that satisfy complex constraints, even in novel scenarios.
- We demonstrate an effective 3D collision avoidance method with lightweight scene features extracted around sampled trajectories and bounding box estimates of the body.
- Our approach broadly applies to the various interaction-rich scenes requiring precise control to generate natural full-body motions.

## 2. Related Work

### 2.1. Human Motion Generation

Recent progress in data-driven approaches for generative models has witnessed remarkable advancements in human motion generation. In addition to the quality and naturalness, many practical applications require generating motions adaptive to diverse conditions. For example, several works allow user to define the input conditions for motion synthesis, such as text [12, 13, 32, 39, 40, 53, 63, 65, 67], music [30, 42, 46, 54] or paired object trajectories [3, 10, 28, 29, 61].

We focus on generating human motions fulfilling practical tasks requiring interaction with diverse geometric layouts. Previous works have long considered motion synthesis in 3D environments. They investigate methods to find plausible root trajectories and complete motions that perform atomic actions such as sitting, walking, and lying [15, 31, 36, 36, 44, 55–57, 69, 70]. Many works mainly consider extracting collision-free paths against cluttered environments. Some frameworks utilize space occupancy [34] or physics simulation [2, 27, 37, 60, 64] to avoid artifacts like penetration, but it is only applicable to a certain range of simple geometries.

More recently, another line of works attempts to generate natural full-body motion especially when grasping an object [49–52]. However, acquiring motion data is challenging in such scenarios, since it is hard to capture the detailed body movements and the paired objects concurrently. Therefore, previous attempts with existing grasping datasets are prone to generate only a limited range of samples due to the insufficient number of reference motions.

Our method especially focuses on generating a human motion that requires a precise goal position for the specific set of body segments. For example, CIRCLE [1] dataset contains various full-body motions reaching for objects in complex spaces. More datasets contain tasks requiring sophisticated controls, such as climbing [62], sitting with provided contact points against chair [68], and motion with contact points with pre-scanned scene [19]. However, the datasets cannot extensively cover intervened constraints in real-world environments.

### 2.2. Diffusion Models and Controllability

Due to the capability to model complex distribution, diffusion-based techniques have demonstrated exceptional performance for generative modeling [7, 16–18]. Motion generation can also benefit from the flexibility of diffusion models that allow sophisticated control of the distribution. Some works [45, 58] employ inpainting techniques to generate motion given joint trajectories, while others [23, 43] proposes a diffusion structure that can modify motion based on root trajectories. AGROL [9] demonstrates a diffusion-

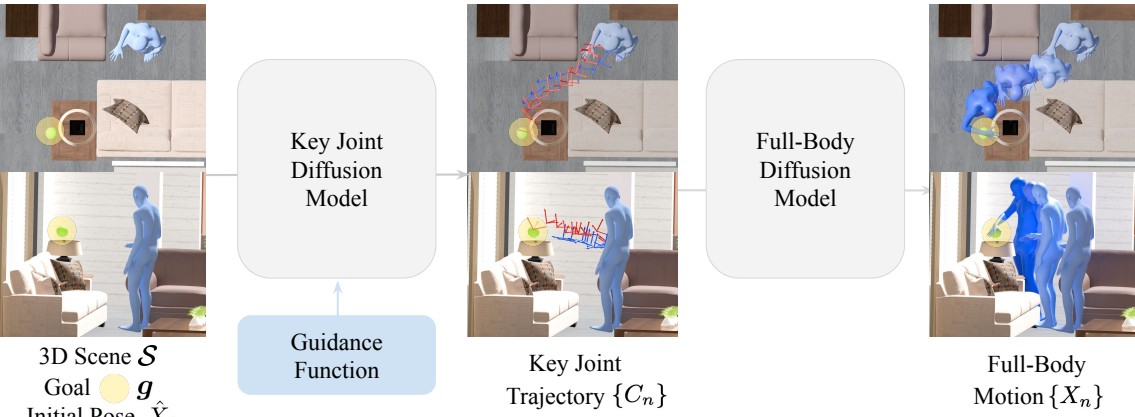

Figure 2. Overall pipeline of our method. Given a 3D scene $\mathcal{S}$ with a set of goal positions $\boldsymbol{g}$ and initial pose $\hat{X}_1$, our goal is to generate smooth and natural full-body motion that reaches the specified goal. We first sample key joint trajectories $\{C_n\}$ satisfying goal conditions using guidance sampling on a diffusion model. Then we feed key joint trajectories $\{C_n\}$ into full-body diffusion model and finally obtain full-body motion $\{X_n\}$.

based framework that reconstructs full-body motion from tracking signals of sparse wearable sensors. Because our task requires creating and matching the desired joint trajectory in an unseen environment, we could also benefit from the flexibility of diffusion models to control the distribution.

We incorporate recent formulations for conditional diffusion to enhance the control for the user-defined task further. ControlNet [66] architecture has emerged as a powerful framework for modeling and sampling high-dimensional data distributions conditioned on input variables. It proposes an additional neural network designed specifically to control image diffusion models such that the results adapt to task-specific control signals. OmniControl [59] pioneered using the ControlNet architecture to generate full-body motion given pre-defined joint trajectory. OMOMO [29] generates hand and body movements step-by-step based on the motion of objects using conditional diffusion formulation. Our work further provides intuitive yet flexible control as the system automatically finds plausible key-joint traces in more challenging environments.

Another way to control the output of a diffusion model is leveraging guidance functions or guided loss functions for flexible sampling [11, 21, 22, 24, 48]. One can use a differentiable loss function to define the necessary constraints for the sampled results. Then, injecting the gradient of this loss steers the output towards the desired form, generating flexible and controllable results. Leveraging guidance and prior knowledge from pre-trained diffusion models, research has made strides in solving linear inverse problems with loss functions akin to the square form [4], or handling non-linear generic loss functions [5]. Recent approaches [47] improve the accuracy of gradients by utilizing multiple Monte Carlo samples to estimate, thereby achieving a more precise ap-

proximation of the gradient. Works such as NIFTY [26] demonstrate that guidance functions can generate more accurate motion. However, such approaches only find the root trajectory with a single object and do not achieve the delicate level of control we propose. By combining conditional diffusion modeling with ControlNet architecture and flexible sampling techniques, and by structuring a two-stage diffusion model, our proposed approach facilitates the generation of natural motions with fine-grained spatial control.

## 3. Method

Given an initial pose of a human $\hat{X}_1$ and 3D goal positions indicated $\boldsymbol{g}$ within a space $\mathcal{S}$, our objective is to generate a sequence of full-body poses $\{X_n\}$ that eventually reach the specified goal positions $\boldsymbol{g}$. Key joints are manually selected for each task, and goal positions are assigned per episode to indicate the task-specific objective. The significant challenge here is to generate plausible and natural motions that satisfy goal conditions while avoiding collisions with surroundings at the same time.

To mitigate these complexities, we propose a two-stage diffusion-based framework. Our framework employs a hierarchical structure that initially generates key joint trajectories $\{C_n\}$ adhering to scene constraints, followed by the creation of full-body motion $\{X_n\}$ based on these trajectories. In addition to the start and end positions of the key joint trajectories, our diffusion process provides a guidance about the potential scene obstructions by encoding the local free space and approximate body configurations given the key joint positions. Based on our lightweight scene features, our model in the first stage finds the 6-DoF paths for the key joint trajectory that effectively avoids collision against cluttered scenes while smoothly approaching the

goal. Then, the next stage can complete a full-body sequence with frame-wise assistance of the key joint trajectory. Our entire pipeline is shown in Figure 2.

**Data Representation** We select $K$ joints from the total joint set and compose our key joint trajectories $\{C_n\} \in \mathbb{R}^{N \times d}$, where $N$ denotes the length of the generated motion sequence. These trajectories $C_n$ contain global $xyz$ position and global 6D rotation [71] of selected key joints, making $d = K \times 9$. For example, if we choose hands and feet for the key joints, then $d = 36$. This global representation enables more direct gradient calculation with spatial constraint-based guidance in Stage 1, without any additional computation, resulting in more accurate sampling [47].

Our full-body motion representation $\{X_n\}_{n=1}^{N}$ includes $N$ full-body poses $X_n \in \mathbb{R}^D$, where $D$ represents the dimension of human pose representation. For the object-reaching scenario and the sitting with contact points task, which involves walking motions, we utilized the HumanML3D [12] representation by converting the root information into global coordinates, following the approach in [23], where $D = 263$. For tasks requiring more natural transitions, such as climbing and contact-aware motion generation, we leverage the parametric human model, SMPL [35], to reconstruct the human mesh at the end of the generation process. The pose vector $X_n \in \mathbb{R}^D$ contains 6 DoF pose of all the joints $J$ and global root translation, where rotations are represented as 6D vectors [71], therefore $D = J \times 6 + 3$.

### 3.1. Stage 1: Key Joint Diffusion Model

Stage 1 generates key joint trajectories that is conditioned on the body shape of the character and the 3D scene layout. A typical denoising diffusion model $\mathcal{D}_\theta$ depends on time $t$ and the additional conditioning feature $\boldsymbol{c}$ in the input data. We employ a network architecture based on U-Net, which learns to recursively sample to recover the original data distribution $p_0(x_0)$ from a noisy version $\boldsymbol{x}_t = \boldsymbol{x}_0 + \sigma_t \epsilon$ with $\epsilon \sim \mathcal{N}(0, I)$. Plugging our formulation into the diffusion model $\mathcal{D}_\theta$, the generated sample $\boldsymbol{x}$ corresponds to the sequence of key joint locations $\{C_n\}_{n=1}^{N}$ and the input condition $\boldsymbol{c}$ is the SMPL shape parameter $\beta$ and the scene $\mathcal{S}$.

#### 3.1.1 Guidance Function

Our diffusion process employs guidance functions to generate samples that precisely satisfy the given goal conditions while avoiding collisions in complex environments. While sampling from naïve diffusion model may not flexibly adapt to novel conditions, we introduce two guidance functions to assist the sampling process (Figure 3): trajectory-control

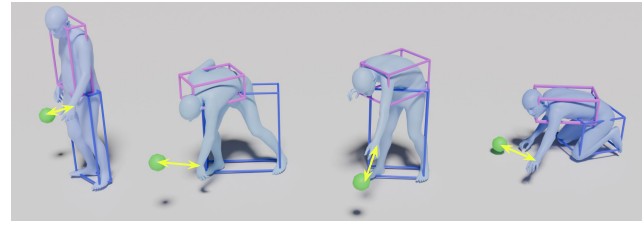

Figure 3. Illustration of guidance functions. We measure the distance between the goal position and corresponding joint for *Trajectory-Control Guidance*. Also, we approximate the body model into a union of the upper and lower body and calculate *Collision-Avoidance Guidance*.

and collision-avoidance guidance. More details on the diffusion process and the calculation of guidance can be found in the preliminary section of the supplementary material.

**Trajectory-Control Guidance** Trajectory-control guidance ensures the generated key joint trajectory smoothly interpolates between the start and the goal position. We formulate the start and the goal guidance, respectively. The start guidance is

$$G_{\text{start}}(\{C_n\}, \hat{X}_1) = \sum_{k=1}^{K} \left\| \mathcal{T}_k(C_1) - \mathcal{T}_k(\hat{X}_1) \right\|_2, \quad (1)$$

where $\mathcal{T}_k(\cdot)$ is the operation to retrieve global $xyz$ position and 6D rotation of a $k$-th key joint from the input vector. The guidance calculates the pose deviation of key joints in the initial frame to ensure starting from the specified initial pose. In a similar context, the goal guidance encourages the model to generate plausible trajectories regarding the goal condition $\boldsymbol{g} \in \mathbb{R}^{K \times 3}$ as following

$$G_{\text{goal}}(\{C_n\}, \boldsymbol{g}) = \sum_{k=1}^{K} \|\mathcal{P}_k(C_N) - \boldsymbol{g}_k\|_2, \quad (2)$$

where $\mathcal{P}_k(\cdot)$ is operation to retrieve global $xyz$ position of $k$-th key joint from the data. Applying the two guidance functions, our diffusion model can generate key-joint trajectories that precisely match the user-defined positions.

**Collision-Avoidance Guidance** In order to prevent potential collisions within the final generated motion, Collision-Avoidance guidance is applied to assist the key joint trajectory $\{C_n\}$. To generate collision-free full-body movement, the guidance has to foresee the entire body movement induced from the key joint configurations in relation to the 3D scene. We provide a guidance by testing collision on points sampled from a geometric proxy of the body volume. Given the canonicalized key joint locations and 6DoF pose in each frame and the body shape parameter

$\beta$, we train a two-layer MLP architecture that estimates the parameters of two bounding boxes, each covering the upper and lower body, as shown in Figure 3. Then, we sample a set of points $\{v\} \in V$ from the estimated geometries and penalize if a point $v$ incurs collision against the surrounding scene $\mathcal{S}$. We identify the possible collision using the signed distance field (SDF) $\Phi_{\mathcal{S}}(\cdot)$ of the scene, by measuring the value at the queried points $V$. As a result, the guidance function is written by

$$G_{\text{collision}}(\{C_n\}, \mathcal{S}, \beta) = -\sum_{v \in V} \mathbb{1}(\Phi_{\mathcal{S}}(v) < 0), \quad (3)$$

where $\mathbb{1}$ is 1 if $\Phi_{\mathcal{S}}(v)$ is negative, i.e., colliding with the scene, and 0 otherwise.

In summary, our final guidance function is defined as a weighted sum of aforementioned guidance terms $\lambda_1 G_{\text{start}} + \lambda_2 G_{\text{goal}} + \lambda_3 G_{\text{collision}}$.

### 3.1.2 Suggestive-Path Feature

We optionally use the suggestive-path feature $\Psi_k$ for a hand trajectory of the task of reaching an object (Task 1 in Sec. 4). In this case, Stage 1 needs planning to find a trajectory within the cluttered scene. The suggestive-path feature is designed to provide a reference trajectory for the end-effector and the scene information around it.

Given the initial pose $\hat{X}_1$ and the goal position $g_j$, we first find a collision-free path of the end effector using the path-finding algorithm [14] within the scene $\mathcal{S}$. Then, we compute geometric features along the path. Specifically, we sample points on the extracted path at regular intervals and extract basis point set (BPS) [41] features, estimating the amount of free space. We concatenate the calculated path with the BPS features computed along the path to derive the suggestive-path features $\Psi_k$ for $k$-th key joint. These features are lightweight yet capable of observing the local scene context, enabling general adaptability. When using this feature, we build an additional feature encoder into our network inspired by ControlNet [66].

### 3.2. Stage 2: Full-Body Diffusion Model

In the second stage, we generate full-body poses $\{X_n\}$ from the trajectory of key joints $\{C_n\}$ and body shape parameters $\beta$. We train another conditional diffusion model, where the condition is given as frame-wise key-joint positions generated from the previous stage. The key joints provide detailed guidance, which already takes the scene context and the goal conditions into account, and Stage 2 can only focus on generating proper full-body motions following the trajectory. Our network architecture integrates the ControlNet [66] structure into the U-Net architecture proposed in [23].

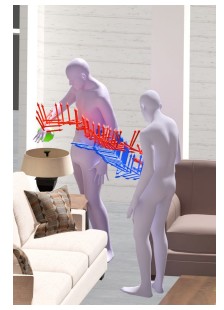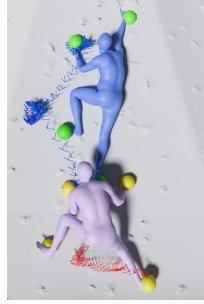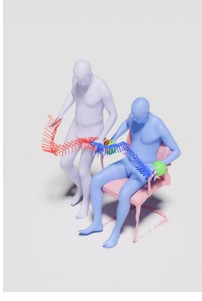

Figure 4. We visualize selected key joint trajectories (blue, red coordinates) from Stage 1, and overlay with the initial and last full-body pose generated from Stage 2. We visualize only a subset of selected key joints for better visualization. Our method successfully synthesizes plausible motions that match the goal conditions as well as the scene context.

## 4. Experiments

Given the initial pose and a 3D scene, our pipeline generates full-body motion that avoids collision and reaches the goal positions for the pre-defined set of key joints of the task. All motion sequences are sampled at 30 FPS. We implement our pipeline using PyTorch [38]. We use the Adam optimizer [25] with a learning rate of $10^{-4}$ for all the experiments. Training requires approximately 24 hours on a single NVIDIA RTX 3090 GPU to cover both Stage 1 and 2. Further details including model architecture and hyperparameters are available in the supplementary material.

We provide a set of metrics to assess the success of the task, physical plausibility, and similarity to the ground truth motion.

- **Success of the task.** At a high level, the task succeeds when a character reaches the goal position without collisions. The *Success rate* indicates that (1) the final position of the key joint is within a predefined distance threshold from the specified goal, and (2) the maximum collision between the generated body model and the scene is within 4 cm. We also calculate the average *Distance to the goal*.
- **Physical plausibility.** For each time step $n$, we calculate the maximum collision distance between the human mesh model from $X_n$ and the given 3D scene $\mathcal{S}$. If this distance exceeds 5 cm, we consider that the collision occurred at the frame. Then, we report the ratio of frames with collisions out of all generated frames as the *Collision rate*.
- **Motion quality.** We assess the motion quality by similarity to the ground truth motion. *Frechet Inception Distance (FID)* evaluate overall motion quality by measuring the distributional distance between ground truth motions and generated motions on the test set. We use four kinds of distance-based metrics to evaluate the difference from the ground truth test data. *HandJPE* quantifies the mean hand joint position errors. *MJPE* is the mean joint po-

| | Method | FID ↓ | Success rate (%) | Dist. to goal (cm) | Collision (%) | Hand JPE (cm) | MJPE (cm) | Root trans. error (cm) |
|---|---|---|---|---|---|---|---|---|
| Random | CIRCLE [1] | 0.338 | 67.06 | 7.97 | 11.77 | **12.93** | **8.03** | 13.15 |
| | OmniControl [59] | 0.372 | 62.40 | 8.03 | 19.43 | 15.84 | 10.59 | 12.09 |
| | Ours single-stage | 0.391 | 61.55 | 7.55 | 23.81 | 16.05 | 11.57 | 13.54 |
| | Ours w/o collision | 0.355 | 56.16 | **7.09** | 26.16 | 20.70 | 12.18 | 16.97 |
| | Ours w/o feature | 0.331 | 66.28 | 7.63 | 13.88 | 15.68 | 9.57 | 11.56 |
| | Ours | **0.319** | **69.07** | 7.22 | **11.62** | 13.24 | 8.39 | **10.38** |

Table 1. Quantitative evaluation on the *reaching an object* scenario. The diffusion network is trained with random splits for the training and the test data.

| Method | Success rate (%) | Dist. to goal (cm) | MJPE (cm) | Root trans. error (cm) |
|---|---|---|---|---|
| OmniControl [59] | 32.2 | 30.05 | 25.54 | 26.41 |
| Ours single-stage | 16.1 | 47.31 | 29.27 | **24.88** |
| Ours | **54.8** | **21.21** | **23.89** | 27.18 |

Table 2. Quantitative evaluation on the *rock-climbing* scenario.

| Method | Dist. to goal (cm) | MJPE (cm) | Root trans. error (cm) |
|---|---|---|---|
| OmniControl [59] | 15.38 | 14.90 | 12.57 |
| Ours single-stage | 21.58 | 19.66 | 25.08 |
| Ours | **14.11** | **13.88** | **10.55** |

Table 3. Quantitative evaluation on the *contact-aware motion generation* scenario.

sition errors in centimeters. We also compute the *Root translation error* using Euclidean distance, measured in centimeters.

To demonstrate the applicability of our motion generation approach, we show successful motion generation on several goal-driven interaction tasks (Figure 1). While the training set-up and constraints vary for different tasks, our two-stage pipeline finds plausible key joint trajectories followed by the natural full-body motion (Figure 4). We provide additional tasks and further task details on supplementary materials.

**Task 1: Reaching an Object Goal in a Cluttered Indoor Scene**   The first task includes the indoor scenes, where the objective is to avoid collisions against the environment while right-hand reaches a specific goal location. Specifically, the right wrist should be within 10 cm of the specified goal to be counted as a success. We designate the *root* and *right hand* as the set of key joints. This scenario is trained with the CIRCLE dataset, which contains 3138 sequences for the task with diverse scene layouts.

We use the algorithm in CIRCLE [1] as a baseline for the experiments. The quantitative evaluations are summarized in Table 1. The training and test datasets are chosen randomly regardless of the scene types in the dataset, and our approach outperforms the baseline in terms of Success rate.

**Task 2: Rock Climbing Guided by Multiple Goals**   As a second task, we show performance on a climbing scenario using the dataset of CIMI4D [62], where multiple key joint goal positions are provided. Here, the task is to generate plausible climbing motions that satisfy multiple positional goals simultaneously. We designate *both feet*, and *hands* as

the key joint set. The success is defined by the positions of both hands and feet at the start and end frames being within 20 cm of the designated rock location. Note that there are eight locations for initial and final conditions to succeed in the task.

The dataset contains only 156 sequences, and we use 125 sequences for training. The task demonstrates that our pipeline can adapt to complex scene constraints and generate natural motion with a limited amount of motion data. Since the 3D scenes in the dataset do not contain clutters with narrow passages, we did not use the suggestive-path features in this task. Table 2 compares our two-stage formulation against a variation employing single-stage generation. Our two-stage pipeline demonstrates superior results in terms of success rate and distance to goals. Due to the lack of sufficient test data to compare distributions, we did not report the FID score. Instead, we visualize overall motion quality in the supplementary videos.

Note that CIRCLE cannot perform the climbing task to reach multiple goals simultaneously because of its initialization scheme. CIRCLE first translates the given initial human body to align with a specific goal point, allowing only a single goal, and subsequently refines the motion. In contrast, our Stage 1 effectively accommodates constraints on multiple key joints that can constitute a unified full-body motion.

**Task 3: Contact-Aware Motion Generation**   We demonstrate that our pipeline can generate motion when extra conditions for intermediate frames are provided. The dataset [19] includes the human motion along with the vertices-level contact, we convert it into joint-level contact using the human body segmentation [35]. For the joints

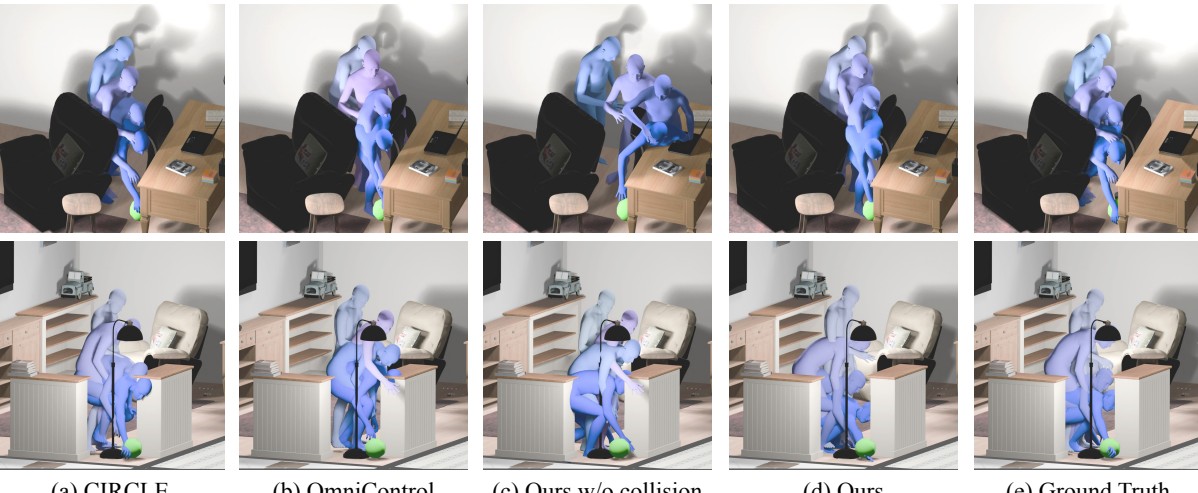

|  (a) CIRCLE | (b) OmniControl | (c) Ours w/o collision | (d) Ours | (e) Ground Truth |

Figure 5. Qualitative results on the *reaching an object*, in unseen scenes with different views. Our method faithfully adapts to the unseen scene geometry in various episodes compared to the presented baselines.

designated as contact joints, we set their global positions as conditions, and our goal is to generate motion while satisfying these conditions. Unlike other tasks, these conditions are also specified for the intermediate frames. Since contacts typically occur at the end-effectors, we designate both *feet* and *hands* as the key joint set. Further details on the processing steps are provided in the supplementary materials. Table 3 shows that our full pipeline outperforms the one-stage pipeline across most metrics. We report the average distance between multiple intermediate goals instead of the Success rate. Our pipeline can also successfully handle multiple intermediate goals.

## 4.1. Efficacy of Detaching the Key-Joint Trajectory

In diffusion models, guidance sampling helps to meet specific conditions, but adding additional gradients to the samples can lead to unnatural results that deviate from the distribution. In single-stage models, guidance is directly applied during the motion generation process, which can reduce the overall quality of the motion. In contrast, our two-stage approach applies guidance in Stage 1 which generates key joints trajectories only, then completes the motion based on Stage 2. This allows us to generate more natural motion by avoiding direct guidance during the motion generation phase while still satisfying the conditions.

We compare the results with a single-stage version of ours and OmniControl [59] which generates the full-body motion directly. To provide similar guidance, we directly extract key joint positions from the full-body motion and calculate trajectory-control guidance compared to the specified goal. For collision-avoidance guidance, we sample points on the surface of the full-body mesh model instead of approximated body geometries similar to [20].

The result from the single-stage model demonstrates the

efficacy of our two-stage design. The results support that our key joint movement successfully extracts valid key joint trajectories that can incur natural full-body motion. Our Stage 1 ensures generating plausible key joint trajectories that guide natural movement for the full body in the subsequent stage. The single-stage diffusion model could produce motions that satisfy the given conditions using guidance sampling, however, it often generates unnatural motion, as visualized in video results. The errors measured with respect to ground truth motion (MJPE, Root trans. error) indicate that the generated movements agree with the captured movement in our outcome.

The advantage of designing a two-stage model is more pronounced when tested with a scarce dataset such as our second task (climbing). In Table 2, the single-stage diffusion model suffers from limited data to express full-body motion and severely overfits and struggles to effectively satisfy unseen conditions composed of multiple goals. In contrast, the key joint diffusion model in Stage 1 can generalize with fewer data as we decompose complex full-body motion distribution into models with lower complexity.

Further, we report the inference speed of our method, and baseline methods in Table 5. Since we compute guidance in stage 1 which is a lightweight 100-step diffusion model, our two-stage diffusion approach achieves faster sampling compared to single-stage diffusion models that calculate guidance for the entire model in the final motion generation phase. Note that CIRCLE [1] is a feed-forward network and handles only single-goal tasks, like Task 1.

## 4.2. Adaptation to Unseen Conditions

Our diffusion framework can adapt to a novel scene and can generalize interaction motions beyond the captured setup. Table 4 contains results that deliberately use different scene

|  | Method | FID ↓ | Success rate (%) | Dist. to goal (cm) | Collision (%) | Hand JPE (cm) | MJPE (cm) | Root trans. error (cm) |
|---|---|---|---|---|---|---|---|---|
| Scene | CIRCLE [1] | 0.471 | 49.75 | 10.72 | 16.31 | **14.23** | **10.32** | 13.84 |
| | OmniControl [59] | 0.394 | 61.13 | 8.49 | 27.43 | 17.52 | 13.02 | 14.88 |
| | Ours single-stage | 0.423 | 58.72 | 9.14 | 28.14 | 19.57 | 13.91 | 14.28 |
| | Ours w/o collision | 0.371 | 52.50 | **7.94** | 31.42 | 22.61 | 14.84 | 16.39 |
| | Ours w/o feature | 0.359 | 62.16 | 8.82 | 15.21 | 16.52 | 13.78 | 14.36 |
| | Ours | **0.341** | **66.41** | 8.34 | **14.21** | 15.15 | 12.86 | **13.32** |

Table 4. Quantitative evaluation on the *reaching an object* scenario tested in novel scenes. We used different scene types for the training and test data split.

| Method | CIRCLE [1] | Ours | Ours single-stage | OmniControl [59] |
|---|---|---|---|---|
| Time (s) | 0.28 ± 0.02 | 28.32 ± 0.39 | 52.90 ± 0.57 | 143.74 ± 0.71 |

Table 5. Inference time comparision with baselines.

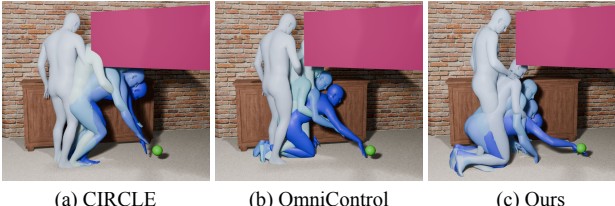

    (a) CIRCLE      (b) OmniControl      (c) Ours

Figure 6. We intentionally added additional obstacles with pink color, and the model demonstrates the ability to generate motions reaching a goal while avoiding collision effectively, even in unseen environments.

types for the training and test split, demonstrating the ability to adapt to different scenes during the test time. Compared to the conventional setup in Table 1, the performance gap is more prominent compared to baseline methods. The scene feature encoding of CIRCLE contains the whole scene from the start to the goal during the entire movement. However, this feed-forward approach performs well only when the scene geometry is similar to those used in training and does not effectively transfer to different geometry. In contrast, our method focuses on localized geometry and performs flexible sampling to meet the conditions within the learned distribution, leading to improved adaptability to novel scene geometries.

We also implement and compare against two-stage versions without collision guidance or suggestive-path features. Motions without collision guidance deteriorate in most quantitative measures, indicating that the term is critical in generating more physically plausible movement within the scene and leading to meaningful improvements in task success rates. The ablation on our suggestive-path feature shows that the feature is effective in increasing the success rates.

Figure 5 shows qualitative results on the generated motion sequences with challenging clutters. Starting from the initial pose, the task is to generate a motion sequence reaching the green dot with the right hand. CIRCLE reaches the target position but cannot refine the motion in the complex scene geometry, resulting in collisions. OmniControl or our diffusion framework with a single stage is insufficient and fails to consider the local geometric context or accomplish the target task correctly. With the proposed guidance, our two-stage pipeline can resolve the challenging task and generate a smooth full-body motion. Figure 6 demonstrates that our generated motions adapt well to new environments or obstacles, aided by collision avoidance guidance with a two-stage pipeline.

## 5. Conclusions

In summary, we introduce a novel approach to generate a goal-driven human motion. Generating motion under pre-defined target positions for specific body joints enables in-tuitive motion synthesis and precise control over character animation. Our two-stage framework can handle a complex goal-driven scenario by solving simpler sub-problems. Especially in cluttered scenarios, our collision avoidance guidance and lightweight scene interaction features facilitate the generation of scene-aware motion. We demonstrate the performance of our pipeline in diverse scenarios, including cases that require rich interaction with multiple goals. Because our model is capable of flexible sampling with minimal data, our pipeline can synthesize natural goal-driven motion even with a limited amount of data.

**Limitations and Future Works** Since the datasets we used do not provide detailed hand motions, our model lacks sophisticated interactions such as grasping objects or navigating climbing rocks. A potential research direction is in the integration of kinematic body motion priors and hand-object interaction priors [2] learned through physics simulators. Also, our method includes task-specific designs, such as manually chosen key joints or toggled features, which are effective for individual tasks but limit its scalability to diverse tasks. This design choice reflects the unique characteristics and requirements of each task and dataset, while the development of a more generalized framework is left as future work.

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
