# OpenReview forum: "Goal-Driven Human Motion Synthesis in Diverse Tasks"
_thecvf.com/CVPR/2025/Workshop/HuMoGen — CVPR 2025 Workshop HuMoGen Submission_

### Official Review · Reviewer_JHzD · 2025-03-26
**A Well-Structured and Controllable Framework for Goal-Driven Human Motion Synthesis**

**Rating:** 5
**Confidence:** 4

**Review:**

This paper presents a two-stage diffusion-based framework for goal-driven human motion synthesis in 3D environments. The pipeline takes as input a 3D scene, an initial human pose, and user-defined key joint goals (e.g., hand or foot target positions), and generates realistic full-body motion that adapts to the scene and satisfies the specified positional constraints.

The proposed two-stage design is both well-structured and practical, effectively decoupling key joint trajectory planning from full-body motion generation. In particular, the key joint trajectory generation guided by goal and collision-aware objectives is a notable contribution that introduces a novel and intuitive mechanism for controlling motion—one that is likely to inspire future research in goal-driven motion synthesis.

Extensive experiments demonstrate the method’s strong generalization, robustness under limited training data, and superior performance compared to existing baselines such as OmniControl and CIRCLE.

One limitation lies in the manual selection of key joints, which may affect the framework's scalability to a broader range of tasks. A promising future direction would be to automate this selection process, for example, by leveraging large language models (LLMs) to infer relevant joints from task and scene descriptions.

---

### Official Review · Reviewer_adEY · 2025-03-26
**The proposed Method is well motivated. It solves a challenging problem and shows applications in multiple tasks. Evaluations are thorough. Visual results are convincing.**

**Rating:** 4
**Confidence:** 4

**Review:**

Strengths:
1. The proposed method is well motivated. The proposed goal driven motion synthesis approach has quite a bit of potential for driving a character through unseen scenarios, and can be applied for various challenging tasks as demonstrated in this paper.
2.	The paper is well written and organized. Literature review is fairly thorough.
3.	The proposed two stage approach is justified through extensive experiments and ablations. The visual results are convincing. The method generates state-of-the-art results for multiple tasks such as various reaching out motions, and even difficult scenarios such as rock climbing .

Weakness:

1.	I am wondering how you choose the correct key joints for a task ? How much does the chosen key joints affect the quality of motion generation? E.g. for reaching out motion, is the key joint always the hand joints or is it possible to also use the shoulder or elbow joints as guidance and then determine the end effectors. I think  it might be interesting to see how the quality of motion changes with the number of key joints specific for the different tasks.
2.	The proposed two stage approach is similar in many ways to the paper: Rempe, Davis, et al. "Trace and pace: Controllable pedestrian animation via guided trajectory diffusion." in the sense that the first stage generates a trajectory and the second stage produces a motion. It would be nice to hear some comments from the authors in which ways these two methods differ in terms of one another.

---

### Meta-Review · Area_Chair_wHMg · 2025-03-31

**Recommendation:** Accept

**Metareview:**

The paper received positive reviews from both the reviewers. The method is sound, the evaluations are thorough and the quality of the results is good. The AC agrees with the reviews and recommends accepting the draft. The authors are encouraged to include the recommendations of the reviewers in the camera ready.

---

### Decision · Program_Chairs · 2025-03-31

Accept